# Synergistic Effects of Nitrogen Application Enhance Drought Resistance in *Machilus yunnanensis* Seedlings

**DOI:** 10.3390/plants14203194

**Published:** 2025-10-17

**Authors:** Jiawei Zhou, Mei Luo, Peng Ning, Songyin Gong, Xiaomao Cheng, Xiaoxia Huang

**Affiliations:** Southwest Landscape Architecture Engineering Research Center of National Forestry and Grassland, College of Landscape Architecture and Horticulture Sciences, Southwest Forestry University, Kunming 650224, China; a1154244478@163.com (J.Z.); 15121532581@163.com (M.L.); np2415406059@163.com (P.N.); 19984061525@163.com (S.G.); xmcheng@swfu.edu.cn (X.C.)

**Keywords:** drought stress, nitrogen forms, nitrogen use efficiency, cultivation management

## Abstract

Drought poses a severe challenge to ornamental tree growth under climate change. This study employed a 2 × 4 factorial design—with two soil moisture levels (80–85% vs. 50–55% field capacity) and four nitrogen treatments (NN: no nitrogen; NO: nitrate nitrogen; NH: ammonium nitrogen; MN: mixed nitrate-ammonium nitrogen)—to examine the efficacy of nitrogen addition in enhancing drought resistance in *Machilus yunnanensis* seedlings. Results revealed that (1) drought stress leads to the acidification of rhizosphere soil, resulting in a decrease of 7.67%, 29.51%, 14.07%, and 44.09% in the content of soil organic matter (SOM), available phosphorus (AP), available potassium (AK), and dissolved organic nitrogen (DON), respectively. This adverse change directly impacts plant growth; it is manifested by a significant reduction of 45% in total chlorophyll (T Chl), a 67.18% decrease in photosynthetic rate (Pn), as well as reductions of 10.61%, 27.59%, 14.81%, and 12.35% in plant height, leaf, stem, and total biomass, respectively. (2) The application of all three forms of nitrogen helps alleviate drought stress, as evidenced by the recovery of photosynthetic levels and the reduction in malondialdehyde (MDA) content, with ammonium-N exhibiting superior efficacy over nitrate-N across most metrics. (3) Strikingly, the mixed nitrogen form outperformed singular applications by demonstrating multifaceted advantages: It maintains soil pH levels and rhizosphere nutrient availability under drought conditions, particularly with a 10.99% and 33.44% increase in dissolved organic nitrogen and available phosphorus content, respectively. More importantly, under drought stress, it increased leaf water content by 20.31%, nitrogen use efficiency by 15.67%, and photosynthetic nitrogen use efficiency by 439.44%, promoted the accumulation of osmolytes, while upregulating antioxidant enzyme activity to counteract osmotic imbalance and alleviate oxidative damage. These findings highlight that nitrogen supplementation, particularly mixed nitrogen application, enhances drought resistance in *M. yunnanensis*, offering a viable management strategy to sustain urban tree landscapes in water-limited environments.

## 1. Introduction

Drought is a key limiting factor for global plant growth, posing serious threats to agriculture, forestry, and urban ecosystems [1,2]. It tends to be neglected that urban plants face enhanced vulnerability to drought due to their unique environment [3]. These plants frequently struggle with poor soil moisture, nutrient deficiencies, and stunted growth. This inevitably diminishes their aesthetic and ecological functions while increasing maintenance costs for urban gardening [4].

The seedling stage is typically the most delicate developmental phase for trees, requiring plants to balance growth with stress resilience under drought conditions [5,6]. For plants, the photosynthetic system is relatively sensitive to water deficiency [7,8]. When water supply fails to compensate for transpiration rates, diminished turgor pressure compromises cellular membrane integrity and triggers reactive oxygen species (ROS) accumulation [9,10]. Excessive ROS buildup then induces stomatal closure via receptor kinase signaling [11,12]. At this stage, stomatal closure along with reduced CO_2_ availability are recognized as primary causes of diminished photosynthetic rate (Pn). Consequently, carbon supply essential for plant metabolism becomes constrained under drought conditions [13,14,15]. Moreover, leaf dehydration causes damage to thylakoid vesiculation and chloroplast membranes, which subsequently reduces chlorophyll content and photosynthetic efficiency. This consequently diminishes carbohydrate accumulation, with seedlings typically manifesting retarded shoot growth, leaf curling, and yellowing [16]. On the other hand, plants respond to drought by adjusting root morphology—for instance, increasing root surface area and length to enhance their ability to acquire water and nutrients under arid conditions [17,18]. However, a decrease in soil water content also affects soil chemical properties, which in turn influences plant uptake of water and nutrients [19]. The decrease in soil water content means a reduction in solvent, which is not conducive to the diffusion of elements such as nitrogen and phosphorus and their absorption by plant roots [20,21]. Therefore, clarifying the differential impacts of drought on plant physiology and rhizosphere environments might facilitate the development of strategies to enhance drought tolerance and seedling survival rates.

Nitrogen utilization is a widely adopted strategy to enhance plant drought resistance in agriculture. Supplementing nitrogen fertilizer not only compensates, to some extent, for the soil nitrogen deficiency caused by drought, but also helps the drought tolerance of plants [22,23]. However, research by Wei et al. also suggests that the roles of distinct nitrogen forms in mediating plant drought tolerance vary substantially [24]. Gorska suggests that NO_3_^−^-N serves as a nutrient and a signaling molecule, and that nitrate contributes to root development and reduces the root-shoot ratio [25]. Furthermore, stomatal regulation and aquaporin activity in plants are dependent on nitrate signaling, and elevated nitrate levels contribute to the maintenance of transpiration under drought conditions [24,26]. NH_4_^+^-N is described as a paradoxical nutrient; NH_4_^+^-N-supplied plants exhibit enhanced root hydraulic conductivity compared to NO_3_^−^-N-supplied counterparts [27,28], yet it concurrently suppresses aquaporin (AQP) activity in plant tissues, impairing drought resistance by hindering water retention and potentially triggering toxicity under prolonged stress [29,30]. Thus, optimizing nitrogen form selection is critical for balancing plant water-use efficiency and drought resilience.

*Machilus yunnanensis*, a native ornamental tree species of Yunnan province, with significant cultural and ecological value, is widely used in urban greening [31]. This study aims to perform the following: (1) Elucidate the effects of drought stress on the physiology of *M. yunnanensis* seedlings and their rhizosphere soil. (2) Investigate how nitrogen application specifically enhances drought resistance in *M. yunnanensis* seedlings through physiological and rhizospheric responses. (3) Determine which nitrogen form (NO_3_^−^, NH_4_^+^, or mixed) is most effective in enhancing drought tolerance. The research seeks to provide theoretical guidance for the urban greening production of *M. yunnanensis* seedlings, including seedling management during the dry season and optimal nitrogen application strategies.

## 2. Results

### 2.1. Effects of Drought and Nitrogen on Physicochemical Properties of Rhizosphere Soil of M. yunnanensis Seedlings

First, we examined how drought affected carbon, nitrogen, and phosphorus (C, N, P) content and their stoichiometry in the rhizosphere soil of *M. yunnanensis* seedlings. Results showed that drought reduced nutrient concentrations in the rhizosphere soil. Specifically, compared with well-watered conditions, total carbon (TC) in the NN treatment decreased by 7.69%, total nitrogen (TN) decreased by 7.30%, and total phosphorus (TP) increased by 3.11% (Table 1). Under drought conditions, nitrogen applications (NO, NH, and MN) significantly increased TN content by 24.54%, 21.98%, and 23.26%, respectively, compared to NN, accompanied by notable stoichiometric shifts. Nitrogen application under drought conditions resulted in a significant decrease in the TC/TN ratio (Table 1) and a significant increase in the TN/TP ratio.

On the other hand, nitrogen supplementation (NO, NH, and MN) can effectively counteract the decline in key soil nutrients caused by drought. In drought conditions, NO treatment increased dissolved organic nitrogen (DON) by 6.36% and available phosphorus (AP) by 19.76%, while NH treatment increased DON and AP by 6.19% and 21.41%, respectively, compared to NN treatment. More prominently, the MN treatment resulted in a 10.08% increase in DON and 33.44% increase in AP. These findings highlight nitrogen application, particularly mixed nitrogen forms, as an effective strategy to enhance soil nutrient availability under drought conditions.

Furthermore, as shown in Table 1, we found that pH level showed minimal response to drought conditions (NN treatment) but exhibited significant modifications in nitrogen treatments (NO, NH, MN treatment, *p* < 0.05). Compared with the NN treatment under drought, nitrate application (NO) elevated pH by 4.52%, while ammonium supplementation (NH) reduced it by 9.77%. Notably, the variation in pH with mixed nitrogen (MN) was smaller than NO or NH treatment.

### 2.2. Drought and Nitrogen Effects on the Growth Morphology of M. yunnanensis Seedlings

To understand the effects of drought on seedling growth, we measured the morphology indices of seedlings. Drought stress inhibited the growth of *M. yunnanensis* seedlings (Table 2, NN treatment), resulting in reductions of 10.61%, 27.59%, 14.81%, and 12.35% in plant height, leaf biomass, stem biomass, and total biomass compared to well-watered conditions, respectively. Notably, drought stress significantly damaged leaf development in the seedlings (*p* < 0.05), with measurements showing that drought treatment reduced single leaf area, total leaf area, and specific leaf area by 25.42%, 43.71%, and 18.49% (Table 2). Furthermore, drought increased the root length of the seedlings by 11.01%, root biomass by 8.00%, and the root-shoot ratio by 33.33%. This indicates that *M. yunnanensis* responds to water scarcity by reducing leaf area and maintaining root growth investment.

However, nitrogen application treatments (NO, NH, and MN) alleviated these drought-induced morphological changes in seedlings, with mixed nitrogen application (MN) proving most effective. Compared to NN-treated plants, the MN treatment resulted in a 39.91% increase in plant height and a 17.11% increase in root length (Table 2). It also significantly (*p* < 0.05) enhanced biomass, with leaf biomass, stem biomass, root biomass, and total biomass increasing by 14.29%, 13.04%, 14.81%, and 14.08%, respectively (Table 2). Additionally, the MN treatment improved leaf morphology, increasing single leaf area by 35.49%, total leaf area by 111.58%, and specific leaf area by 52.44%. These results collectively demonstrate that nitrogen application, particularly mixed nitrogen application, supports better growth maintenance in *M. yunnanensis* seedlings under drought stress.

Moreover, the results of relative water in leaves (LRWC) showed that drought stress significantly reduced leaf relative water content by 28.09% (Table 2, compared to NN treatment, *p* < 0.05). However, nitrogen supplementation effectively mitigated this water deficit. All nitrogen forms (NO, NH, and MN) significantly improved leaf water status under drought (*p* < 0.05). Among them, mixed nitrogen (MN) showed superior water retention capacity—20.31% higher than the NN treatment.

### 2.3. Drought and Nitrogen Effects on Photosynthetic Pigment Contents of M. yunnanensis Seedlings

To clarify how drought stress specifically inhibits the growth of *M. yunnanensis* seedlings, we conducted measurements of the following parameters. We observed that drought reduced the levels of chlorophyll a (Chl a), chlorophyll b (Chl b), and carotenoids (Car) levels by 46.81%, 39.47%, and 26.53%, respectively (Figure 1a–c). This disproportionate decline elevated the Car/Chl ratio by 32.14% in the NN treatment (Figure 1f).

Simultaneously, we observed that nitrogen supplementation (NO, NH, and MN) significantly (*p* < 0.05) enhanced the accumulation of Chl a and Chl b in both conditions. Notably, MN application substantially boosted T Chl concentrations by 121.21% (Figure 1d) and reduced the Car/Chl ratio by 29.73% compared to the NN treatment under drought stress.

### 2.4. Drought and Nitrogen Effects on Photosynthetic Parameters in M. yunnanensis Seedlings

Associated with the decline in photosynthetic pigment content, we observed a corresponding reduction in photosynthetic efficiency in *M. yunnanensis* seedlings. Specifically, under drought stress, the maximum quantum yield of PSII (Fv/Fm, Figure 2a), the actual photochemical efficiency (ΦPSII, Figure 2b), and the photochemical quenching coefficient (qP, Figure 2c) decreased by 15.63%, 48.89%, and 13.33%, respectively. Concurrently, the non-photochemical quenching (NPQ, Figure 2d) parameter increased by 70.45%. Similarly, nitrogen supplementation (NO, NH, and MN) offsets these declines, with mixed nitrogen (MN) demonstrating the most pronounced recovery of photosynthetic function. Under drought, the MN treatment increased Fv/Fm by 40.74%, ΦPSII by 134.78%, and qP by 29.23%, but decreased NPQ by 25.33% compared to the NN treatment. These results indicate that drought inhibits plant photosynthesis, while nitrogen supplementation can partially mitigate this effect.

Further supporting these findings, gas exchange measurements highlighted the adverse effects of drought on photosynthetic performance (Figure 2e–h). Drought caused a significant reduction of 67.18% in net photosynthetic rate (Pn) and 75.58% in transpiration rate (Tr) for the NN treatment (*p* < 0.05). Nitrogen treatments universally reversed these trends, with the MN application yielding the most robust improvements. Compared to the NN treatment, the MN supplementation elevated Pn by 260.43%, stomatal conductance (Gs) by 500.00%, intercellular CO_2_ concentration (Ci) by 84.92%, and Tr by 271.43%, underscoring its critical role in enhancing stomatal regulation and carbon assimilation under drought stress. These findings consistently underscore the role of nitrogen addition—particularly mixed nitrogen application—in enhancing the carbon assimilation capacity of *M. yunnanensis* seedlings.

### 2.5. Drought and Nitrogen Effects on Leaf Nitrogen Use Efficiency in M. yunnanensis Seedlings

The following results further show that drought and nitrogen effects on leaf total carbon (TC_leaf_), total nitrogen (TN_leaf_), nitrogen use efficiency (NUE), and photosynthetic nitrogen use efficiency (PNUE) in *M. yunnanensis* seedlings. Drought significantly reduced TC, TN, NUE, and PNUE (*p* < 0.05). Compared to NN, the NO, NH, and MN treatments increased TC by 33.99%, 40.14%, and 53.81%, respectively (Figure 3a); TN by 18.76%, 27.02%, and 32.99%, respectively (Figure 3b); NUE by 12.92%, 10.35%, and 15.67%, respectively (Figure 3c); and PNUE by 115.49%, 167.61%, and 439.44%, respectively (Figure 3d) under drought stress. It can be inferred that nitrogen application, especially the mixed nitrogen form, can improve the PNUE in *M. yunnanensis* seedlings, thereby supporting their photosynthesis and resistance under drought stress.

### 2.6. Drought and Nitrogen Effects on Osmolytes and Antioxidant Capacity in M. yunnanensis Seedlings

To understand the nature and extent of drought-induced damage in *M. yunnanensis* seedlings and the role of nitrogen application, we measured MDA and osmolyte content in leaves. Key observations revealed drought-induced elevations in leaf proline for 157.84%, soluble protein for 21.21%, soluble sugars for 99.83% (Figure 4a–c), and MDA for 88.97% relative to well-watered conditions (Figure 4d). Nitrogen application under well-watered conditions showed no significant effects on these parameters (*p* > 0.05). Under drought, however, all nitrogen treatments (NO, NH, and MN) increased proline, soluble proteins, and soluble sugars while reducing MDA levels, with mixed nitrogen (MN) yielding maximal responses: 103.14% and 25.13% increases in proline and soluble sugars, 20.47% elevation in soluble proteins, and a 44.21% decrease in MDA compared to NN. On the one hand, nitrogen application significantly elevated osmolyte content under drought stress, on the other hand, it suggests that nitrogen mitigates drought-induced cellular damage. Among these, the performance of the MN treatment in alleviating drought-induced damage is particularly notable.

Meanwhile, we measured the activities of four key antioxidant enzymes (APX, POD, CAT, SOD) to understand how *M. yunnanensis* seedlings respond to reactive oxygen species (ROS) imbalance caused by drought, and whether nitrogen addition could help enhance their antioxidant capacity. Drought stress significantly enhanced antioxidant enzyme activities (*p* < 0.05) in *M. yunnanensis* seedlings (Figure 4e–h). However, compared with the NN treatment, nitrogen application in the forms of NO, NH, and MN increased APX activity by 13.94%, 11.51%, and 52.73%, respectively; POD activity was increased by 10.02%, 22.53%, and 63.85%; CAT activity was increased by 62.25%, 83.68%, and 109.20%; and SOD activity was increased by 5.72%, 9.41%, and 32.66%. We propose that nitrogen addition, especially the MN treatment, enhances the activity of the antioxidant enzyme system in *M. yunnanensis* seedlings under drought conditions, enabling them to counter the risk of excessive ROS accumulation.

### 2.7. Drought and Nitrogen Effects on Rhizosphere Nutrients in Relation to Growth and Drought Resistance of M. yunnanensis Seedlings

Finally, we employed redundancy analysis (RDA) to investigate the responses of multiple parameters in soil and plant physiology under different water conditions, elucidating two dominant factors influencing seedling growth and stress resistance (Figure 5). In addition, correlation analysis (Figure 6) revealed the relationships between rhizosphere and plant physiological indicators under drought and nitrogen application treatments.

The results of the RDA indicate that under well-watered conditions, 78.16% of the variation between the growth characteristics of *M. yunnanensis* seedlings and soil physicochemical properties can be explained by the first two axes (RDA 1 and RDA 2), with DON content contributing to 59.50% of this variation (Figure 5a). Similarly, 64.81% of the variation between stress resistance indicators of *M. yunnanensis* seedlings and soil physicochemical properties can be explained by the first two axes under the same conditions, with 30.40% of this variation explained by DON content (Figure 5b). In drought conditions, 86.29% of the variation between the growth characteristics of *M. yunnanensis* seedlings and soil physicochemical properties can be explained by the first two axes, with 60.10% of this variation attributable to AP content (Figure 5c). Meanwhile, 73.06% of the variation between stress resistance indicators and soil physicochemical properties can be explained by the first two axes (Figure 5d), with 44.40% of this variation explained by AP content (Figure 5d).

In addition, correlation analysis showed that DON and AP contents strongly correlated with enhanced photosynthesis and biomass in both the well-watered and drought conditions (Figure 6). This evidence shows that nitrogen and phosphorus availability is important for plant growth and drought resistance. Conversely, TP negatively impacted leaf nitrogen, carbon, and leaf area, suggesting phosphorus antagonism during drought. TK limitation exacerbated oxidative stress via negative correlations with proline and peroxidase. Notably, the application of ammonium nitrogen (NH) outperforms nitrate nitrogen (NO) in supporting photosynthesis and osmotic regulation, while mixed nitrogen application (MN) yields superior results compared to either applied alone.

In summary, DON plays a dominant role in the growth and stress resistance of M. yunnanensis seedlings under well-watered conditions (Figure 5a,b). Under drought conditions, AP becomes the key driver of growth and stress resistance (Figure 5c,d). Correlation analyses (Figure 6) revealed that DON and AP strongly promoted photosynthesis and biomass across all conditions. Thus, DON and AP are the primary factors influencing seedling physiology under well-watered and drought conditions, respectively.

## 3. Discussion

Application of mixed nitrogen can most effectively counteract the adverse effects of drought on soil properties in the cultivation of *Machilus yunnanensis* seedlings. Relevant studies indicate that soil moisture serves as a pivotal regulator of pH, with elevated moisture levels typically reducing pH. This phenomenon is attributed to the leaching of H^+^ ions from soil by water, while simultaneously stimulating soil microorganisms to release H^+^ ions through nitrification processes [32,33]. Contrary to the prevailing view, we found that drought caused a slight decrease in soil pH in this experiment (Table 1). This discrepancy may be explained by the leaching effect of irrigation water being weaker than rainfall, which reduced water flushing and thus its impact on pH, while the lower soil pH under drought was likely induced by biological processes associated with root exudation [34]. Additionally, since the pH of the experimental irrigation water ranged from 6.83 to 7.29, this could be another reason why the pH level in well-watered treatment conditions was higher than in drought conditions. Consistent with previous studies, nitrate-N application induced marked soil alkalization, whereas ammonium-N treatment caused acidification [35,36,37]. However, mixed-N supply neutralized the adverse effects of nitrogen fertilization on soil pH; the mixed nitrogen application (MN) resulted in a smaller changes (Table 1) in pH compared to the single nitrogen application (NO, NH). It is speculated that during the absorption of nitrate and ammonium nitrogen, plant roots release excess H^+^ and OH^−^ ions into the rhizosphere soil, where these ions with opposite acid-base properties collectively stabilize the pH [38,39,40]. This demonstrates that optimizing nitrogen form ratios is key to avoiding or mitigating its negative impacts on soil acidification.

Soil carbon, nitrogen, and phosphorus concentrations constitute fundamental indicators for evaluating soil quality and nutrient cycling efficiency, the chemical contents, and the relationships between them, providing insights into how nutrients are cycled and utilized by plants in adversity [41,42]. Among them, the content of organic carbon is an important factor controlling the immobilization of NH_4_^+^ and NO_3_^−^ by microorganisms [43,44]. Based on this pot experiment, the primary sources of organic carbon input can be inferred as root exudates and microbial carbon sequestration. Therefore, under drought stress, the decrease in TC content and organic matter content in the soil might inhibit root exudate activity caused by water deficit (Table 1). This physiological constraint reduces microbial biomass, consequently inhibiting the decomposition of organic matter and resulting in insufficient input of carbon content [21,45,46]. Meanwhile, the increase in TC/TP and TN/TP (Table 1) under nitrogen treatment indicates that nitrogen application reduced the total phosphorus content in the soil to some extent, with the greatest decrease observed under MN treatment (Table 1). This may be attributed to the synergistic absorption mechanism between nitrogen and phosphorus, whereby sufficient nitrogen availability prompts plants to accelerate phosphorus uptake from the soil to meet biomass accumulation demands, thus reducing soil phosphorus content [47,48].

Further supporting this perspective, redundancy analysis revealed that under drought conditions, available phosphorus emerged as the primary driver of plant performance, accounting for 60.10% of the variation in growth (Figure 5c) and 44.40% in stress resistance (Figure 5d). Conversely, under well-watered conditions, dissolved organic nitrogen dominated the growth physiology and stress resistance of *M. yunnanensis* seedlings. This demonstrates that the interaction between these two elements facilitates the fulfillment of nutritional requirements for both growth and stress tolerance in *M. yunnanensis* seedlings under drought stress. Under drought conditions, the application of the mixed nitrogen treatment (MN) significantly increased DON and AP levels by 10.99% and 33.44%, respectively, compared to NN treatment. The absorption of these substances is beneficial for root development and maintains aquaporin activity under drought stress, thereby sustaining plant hydraulic conductivity [49,50,51]. The correlation analysis further revealed that under both conditions, the dissolved organic nitrogen content in rhizosphere soil demonstrated a notable positive correlation with TN_leaf_, TC_leaf_, and chlorophyll content, and an extremely significant positive correlation with Pn, thereby reinforcing this viewpoint (Figure 6). Given nitrogen’s essential role in plant protein synthesis and metabolic regulation, this DON accumulation likely facilitates enhanced photosynthetic investment and morphological development [26]. In summary, under drought conditions, mixed nitrogen application effectively counteracts the adverse effects of single nitrogen forms on soil properties and improves the rhizosphere nutrient status of *M. yunnanensis* seedlings.

Nitrogen application mitigates drought-induced developmental impairments in *Machilus yunnanensis* seedlings.

Undoubtedly, drought has exerted adverse effects on the growth of *M. yunnanensis* seedlings. These effects are primarily manifested in significant reductions in leaf water content and biomass. We also observed that nitrogen application treatments partially mitigated the impact of drought. The primary effect was an increase in plant height, single leaf area, and total leaf area. MN treatment outperformed the individual application of the two nitrogen forms, with ammonium nitrogen demonstrating superior effectiveness compared to nitrate nitrogen (Figure 2).

In this study, drought stress induced decreases in plant height, leaf biomass, stem biomass, root biomass, and total biomass, while the root-shoot ratio increased. This is consistent with many previous reports, indicating that plant morphogenesis is negatively affected by drought as normal physiological activities are inhibited [52,53]. We also observed that in response to such conditions, the root length and root biomass of *M. yunnanensis* seedlings still showed a slight increase. This indicates that *M. yunnanensis* tends to maintain its root system level to meet its absorption needs under drought stress [54,55]. Furthermore, the increased leaf length-width ratio and lower specific leaf area reduce potential transpiration, which is beneficial for enhancing the plant’s tolerance to drought conditions [56].

In this experiment, nitrogen application partially alleviated drought-induced constraints on plant growth, promoting resource allocation to aboveground parts and thereby sustaining plant development under drought conditions [57]. Under nitrogen application treatments, the total carbon and total nitrogen content, NUE and PNUE in the leaves of *M. yunnanensis* seedlings all increased, with the most significant increase observed in the case of mixed application of nitrogen. This indicates that all three forms of nitrogen are beneficial to plant growth under drought conditions, and the mixed application of ammonium and nitrate nitrogen has the best effect (Figure 2 and Figure 3). Mixed nitrogen promotes plant growth and yield, as reported in many studies [58,59,60]. One study suggests that NO_3_^−^ is primarily assimilated in leaves, with its uptake promoting cellular accumulation and leaf growth in aboveground parts [61]. In contrast, most NH_4_^+^ is absorbed by roots, which helps reduce the energy costs associated with nitrogen transport within plants [62,63]. Thus, under drought conditions, the interplay between two nitrogen forms enhances efficient allocation and utilization of nitrogen required for growth in *M. yunnanensis* seedlings [58,64].

Furthermore, the correlation heatmap revealed that leaf nitrogen content was significantly correlated with total biomass, specific leaf area, and total chlorophyll content (Appendix A
Figure A1), confirming nitrogen’s pivotal role in enhancing growth and photosynthetic capacity under water limitation.

Nitrogen application, particularly mixed nitrogen form, enhanced photosynthetic performance in *Machilus yunnanensis* seedlings under drought.

Photosynthesis is among the most sensitive processes to drought stress, and the decline in photosynthetic capacity is commonly recognized as a major factor limiting plant biomass accumulation [65]. In this study, the decline in photosynthetic capacity may be caused by many reasons. Primarily, drought stress caused a significant reduction in seedling chlorophyll content, and an inverse correlation was observed between chlorophyll content and MDA levels (Figure 1 and Figure 4, Appendix A
Figure A1). This phenomenon is commonly attributed to the decrease in water content in leaf cells, leading to pressure imbalance in chloroplasts due to water shortage. At this time, the thylakoid structure is damaged, shifting the chlorophyll synthesis-degradation equilibrium toward catabolism [66]. A possible explanation is that the supplementation of the two forms of nitrogen is closely related to Rubisco content, thylakoid structure composition, and chlorophyll synthesis, providing substantial protein materials for this process, which in turn affects the photosynthetic system and photosynthetic rate [67].

In addition, some studies have pointed out that stomatal limitation is the main factor causing a decrease in plant photosynthetic rate during drought. This is because drought induces the synthesis of abscisic acid, which acts as a signaling molecule to induce stomatal closure in leaves. In this study, drought stress not only led to a reduction in stomatal conductance but was also accompanied by a decrease in transpiration rate, which may have further resulted in reduced CO_2_ availability and increased respiratory consumption in the leaves. This is generally considered a manifestation of the plant’s reduced capacity to utilize CO_2_; it might be another reason for the decrease in photosynthetic capacity [12]. However, our study demonstrates that nitrogen supplementation effectively counteracts drought-induced photosynthetic impairment in *M. yunnanensis* seedlings (Figure 2). Under drought stress, the photosynthetic parameters of *M. yunnanensis* seedlings decrease, leading to a reduction in carbon assimilation capacity. However, nitrogen treatment (NO, NH, and MN) increases Pn, Ci, Gs, and Tr. Correlation analysis indicates that total chlorophyll content and total leaf area are positively correlated with Pn, Ci, Gs, and Tr.

Furthermore, Fv/Fm, ΦPSII, and qP increase after nitrogen application, while NPQ decreases. This suggests that the accessibility of the PSII reaction center and light use efficiency in the functional leaves of *M. yunnanensis* seedlings increase, further supporting the perspective that nitrogen application can improve plant physiological activity under drought stress. One possible reason is that sufficient nitrogen supply enhances the carboxylation activity of Rubisco. Therefore, the demand for NADPH and ATP generated by the light reaction in the Calvin cycle increases, restoring the activity of the PSII reaction center and improving photochemical efficiency [68,69]. Another possibility is that sufficient nitrogen level allows the light energy absorbed by light-harvesting pigments to be utilized more effectively during transport and conversion. This leads to a decrease in the proportion of energy released during thermal dissipation, manifested as an increase in qP and a decrease in NPQ (Figure 2c,d), reducing the damage to PS [70]. At this time, the photosynthetic rate increases, stomatal limitation is alleviated [71], and the gas exchange capacity is enhanced, leading to an increase in Ci content. In addition, some research has shown that the addition of nitrogen can protect photosynthetic organs to some extent, which can be attributed to nitrogen stimulating the production of osmolytes [72].

Nitrogen application enhances osmotic regulation and antioxidant enzyme activities in *Machilus yunnanensis* seedlings under drought.

Under drought stress, plants accumulate osmolyte substances in their cells to lower water potential, maintain the cytoderm elasticity, and preserve cellular structure. The accumulation of these osmolytes is beneficial for plants in mitigating xylem cavitation risks under drought conditions and regulating their antioxidant systems [13]. In this study, Proline and soluble sugar increased significantly under drought stress, and proline, soluble protein, and soluble sugar further increased after nitrogen application (Figure 4). Among them, proline is regarded as the primary non-enzymatic antioxidant compound produced under drought stress, where it functions both as an osmoprotectant and a redox-buffering agent, exhibiting antioxidant properties [12,73]. In addition, soluble sugars not only serve as cellular osmoprotectants but also provide energy for protein synthesis [74], and soluble proteins serve as vital scaffolds for the synthesis of biological macromolecules, contributing to the maintenance of stability in cell membranes and protoplasmic colloids [75]. The increase in the levels of these three osmolytes indicates that nitrogen application helps enhance the ability of *M. yunnanensis* seedlings to maintain intercellular osmotic balance under drought stress. This finding is consistent with previous research conducted by Ma et al. on *Fraxinus mandshurica* and Chai et al. on *Punica granatum*, whereby nitrogen availability potentiates osmotic adjustment capacity across woody species under water limitation [76,77].

On the other hand, the content of osmolytes (proline, soluble protein, and soluble sugar) also showed a significant positive correlation with antioxidant enzyme activity (Appendix A
Figure A1). This suggests that *M. yunnanensis* seedlings can mitigate the negative effects of drought stress through the synergistic effects of osmotic content and antioxidant enzyme activity. Under drought conditions, the application of nitrogen further increased the activity of APX, SOD, POD, and CAT in the leaves of *M. yunnanensis* seedlings, and at the same time, reduced the content of MDA. This may be closely tied to the regulatory mechanisms of antioxidant enzymes—as their activity is directly dependent on nitrogen availability and levels [78]. This finding is consistent with the results of Tao et al. on *Gossypium hirsutum* and Zhuang et al. on *Robinia pseudoacacia* seedlings regarding moisture and nitrogen supply. Under drought conditions, nitrogen application can compensate for the effects of water deficit, reducing MDA content and enhancing the activity of the antioxidant enzymes [79,80].

In conclusion, this study suggests that nitrogen application under drought conditions enhances the rhizosphere soil environment and improves drought resistance of *M. yunnanensis* seedlings. The application of mixed nitrogen could avoid soil acidification or alkalization caused by the sole application of nitrate or ammonium nitrogen and enhance the availability of dissolved organic nitrogen and available phosphorus and potassium. Furthermore, applying mixed nitrogen demonstrated significant effects in maintaining leaf water balance, promoting leaf nitrogen accumulation and photosynthetic nitrogen use efficiency, and enhancing osmotic regulation and antioxidant enzyme activity, ultimately resulting in increased biomass for the drought-treated seedlings. Therefore, this strategy can provide a scientific basis for the maintenance of urban greening tree species, optimize tree water-use efficiency, sustain ecological functions, and reduce maintenance costs, thus supporting the sustainable construction of drought-sensitive urban greenery.

## 4. Materials and Methods

### 4.1. Study Site and Climate

The experiment was conducted at the Arboretum of Southwest Forestry University, Kunming, China (24°23′–26°22′ N, 102°10′–103°40′ E), characterized by a low-latitude plateau monsoon climate. The region has an annual average temperature of 14.7 °C, precipitation of 1011.2 mm (86% concentrated in summer and autumn), and a spring drought frequency exceeding 75% [81].

### 4.2. Experimental Design

In July 2020, six-month-old *M. yunnanensis* seedlings were planted in polyethylene pots (3L pot volume, 17.5 cm tall, 3:2 humus-red soil mixture). Healthy, uniformly sized seedlings (plants were about 7 cm in height, with approximately 5 leaves.) were selected in August 2020 for experimental treatments under a 2 × 4 completely randomized design. The experiment involved two soil moisture levels: 80–85% field capacity (W, well-watered) and 50–55% field capacity (D, Drought), four nitrogen treatments were applied in each experimental treatment: no nitrogen (NN), nitrate nitrogen (NO), ammonium nitrogen (NH), and mixed nitrate-ammonium nitrogen (MN), with 10 pots (1 seedling per pot) per replicate, 3 replicates in 1 treatment. Soil moisture was maintained at target levels using the weighing method [82].

### 4.3. Nitrogen Application

The drought-stressed plants, following a two-week period of drought stress, together with the well-watered seedlings, were subjected to different nitrogen fertilizer treatments. Based on Wang’s [83] study, the annual fertilization rate for six-month-old *M. yunnanensis* seedlings was set at 45 kg N·ha^−2^ (0.228 g N·pot^−1^). Considering that the area of the pots was about 0.0079 m^2^, nitrogen was applied as 1.645 g·pot^−1^ KNO_3_ (NO treatment), 0.871 g·pot^−1^ NH_4_Cl (NH treatment), and 0.822 g·pot^−1^ KNO_3_ + 0.435 g·pot^−1^ NH_4_Cl mixture (MN treatment), divided into four equal applications of 0.057 g N·pot^−1^ every 7 days. After accurate weighing, nitrogen compounds were dissolved in the weekly water used to maintain field capacity and applied around the roots. All chemical reagents were obtained from the Shanghai Research Institute of Chemical Industry. All *M. yunnanensis* seedlings were harvested in December 2020 for growth and physiological parameter analysis (plants were around 9–15 cm height, with 5–10 leaves.).

### 4.4. Soil Physicochemical Properties

After removing bulk soil from potted plants, the rhizosphere soil attached to roots was collected by brushing. NO_3_^−^-N and NH_4_^+^-N contents were determined using 10 g of soil sample, respectively, whereas a 50 g sample was allocated for the analysis of other soil physicochemical properties. Each treatment included 3 replicate samples. Total organic carbon (TC), derived from potassium dichromate volumetric oxidation, with soil organic matter (SOM) calculated as SOM = TC × 1.724; and total nitrogen (TN), quantified using the Kjeldahl digestion method.

Nitrate-nitrogen (NO_3_^−^-N) and ammonium-nitrogen (NH_4_^+^-N) were extracted with 50 mL of 2 mol/L potassium chloride (KCl) solution and determined by the cadmium reduction-diazo coupling colorimetric method and indophenol blue colorimetric method, respectively. Dissolved organic nitrogen (DON) is calculated as follows:DON = TN − (NO_3_^−^-N + NH_4_^+^-N)

Total phosphorus (TP) and available phosphorus (AP) contents were determined by the molybdenum antimony anti-colorimetric method. The total potassium (TK) content was measured by the sodium hydroxide fusion method, while the available potassium (AK) content was determined by the ammonium acetate extraction method [84].

### 4.5. Irrigation Water

The irrigation water was from an EU-T high-purity water system (OuKai Environmental Technology Co., Ltd., Nanjing, China). The pH of the irrigation water was determined as 6.83–7.29 by using a 2100 pH meter (Starter, Ohaus Co., Los Angeles, CA, USA).

### 4.6. Growth Parameters

In December 2020, plant height and root length were measured in 3 randomly selected seedlings from every replicate. Plant height and root length were measured using a tape ruler, where root length was determined as the total distance from the main root to the tip. All plants were washed, and parts were divided into roots, stems, and leaves, oven-dried at 80 °C, 48 h to constant weight (MOV-212F Oven, Sanyo, Osaka, Japan). Total biomass and biomass of roots, stems, and leaves were recorded. Measurement of leaf relative water content (LRWC): The leaves were rinsed under a stream of deionized water, then blotted dry with filter paper to remove residual surface moisture. Their fresh weight was measured and recorded. The weighed leaves were then labeled and immersed in deionized water for 24 h to ensure complete saturation. After saturating, the leaf surfaces were again blotted dry with filter paper, and the saturated weight was measured and recorded. Following the saturated weight measurement, the leaves were placed in an oven at 80 °C for 12 h. The measurements were completed by determining the dry weight. The measurements were calculated as follows:LRWC (%) = (Fresh weight − Dry weight)/(Saturated weight − Dry weight) × 100
The root-shoot ratio is expressed as follows:Root-shoot ratio = Root biomass/(Leaf biomass + Stem biomass)

### 4.7. Measurement of Carbon (TC_leaf_) and Nitrogen (TN_leaf_) Content in Leaf

The leaf samples were dried at 80 °C for 36 h, then ground and homogenized into a fine powder using a mechanical grinder, before being passed through a 20-mesh screen. Then, use the C and N elemental analyzer (Vario Macro, Elementar, Langenselbold, Germany) to measure the content of C and N in the samples [85].

### 4.8. Calculation of Nitrogen Use Efficiency (NUE) and Photosynthetic Nitrogen Use Efficiency (PNUE)

NUE is calculated by the ratio of total carbon content to TN content in leaves. PNUE is derived from the ratio of the net photosynthesis rate (Pn, μmol·m^−2^·s^−1^) to the nitrogen content per unit leaf area (N_area_, g·m^−2^). In the formula, DM represents the mass (g) of the leaves after drying to a constant weight, LA stands for leaf area (m^2^), TN_leaf_% denotes the TN content of the leaves (%), and SLW represents specific leaf weight (g·m^−2^) [86].

The calculation formula of NUE is as follows:TC_leaf_/TN_leaf_

The calculation formula of PNUE is as follows:SLW = DM/LAN_area_ = (TN_leaf_% × SLW)/100PNUE = Pn/N_area_

### 4.9. Measurement of Photosynthesis and Chlorophyll Fluorescence

Photosynthetic parameters were measured using Li-6400XT Portable Photosynthesizer (LI-COR Biosciences, Lincoln, NE, USA). Five plants were selected from each replicate for photosynthetic parameters determination, and three leaves of each plant were measured and the average value was taken. The net photosynthetic rate (Pn), stomatal conductance (Gs), intercellular carbon dioxide concentration (Ci), and transpiration rate (Tr) were measured using carbon dioxide cylinders with a concentration setting of 400 μmol·mol^−1^, air flow rate was set at 400 mL/min, light intensity was set at 1200 μmol·m^−2^·s^−1^, leaf temperature was set at 25 °C, and relative humidity was set at 60%.

The fluorescence parameters were measured by Modulating Imaging-Pam Series Chlorophyll Fluorescence Imaging System (Walz, Effeltrich, Germany); the initial fluorescence (F0) was measured using a measuring light (0.5 μmol·m^−2^·s^−1^) after the leaves had been dark-adapted for 30 min. The maximal fluorescence (Fm) was induced by saturated light of pulse 2700 μmol·m^−2^·S^−1^ (pulse time 0.8 s), in which the light intensity was 76 μmol·m^−2^·S^−1^. Chlorophyll fluorescence parameters such as maximum photochemical efficiency of PSII (Fv/Fm), photochemical efficiency of PSII (ΦPSII), non-photochemical quenching coefficient (NPQ) and photochemical quenching coefficient (qP) were recorded.

### 4.10. Measurement of Photosynthetic Pigments, MDA, Osmolytes Content, and Antioxidant Enzyme Activities

From each replicate, 5 plants were selected, from which 3–5 fully expanded top leaves were taken for measurements. A sample of 0.2 g was used for the measurement of each indicator. Photosynthetic pigments were determined using the acetone extraction method. MDA content was quantified by the thiobarbituric acid method [82]. Proline (Pro) content was assessed by the ninhydrin colorimetric method [87]. Soluble protein (SP) content was determined using the Komas Brilliant Blue (G-250) staining method. Peroxidase (POD) activity was measured using the guaiacol method. Superoxide dismutase (SOD) activity was quantified by the NBT method. Catalase (CAT) and ascorbate peroxidase (APX) activities were determined by the UV absorption method [88]. Soluble sugars (SS) were measured by the GOD-POD colorimetric method [89].

### 4.11. Statistical Analysis

Data were analyzed by one-way ANOVA (SPSS 25.0 version, IBM Corporation, Armonk, NY, USA) with Duncan’s test for post hoc comparisons at a significance level of *p* < 0.05. Two way analyses (ANOVA, SPSS 25.0 version) of variance were conducted to evaluate the significance of the effect of drought, nitrogen, and their interaction in each population. Graphical representations were generated using Origin 2021 (9.8.0.200 version, OriginLab Corporation, Northampton, MA, USA). Intervariable correlations were assessed through Pearson’s correlation coefficient analysis implemented via the ‘cor’ function in the ggcorrplot package (v0.1.4.1) within the R environment (v4.2.2). Redundancy analysis (RDA) was conducted using Canoco5 software (5.02 version, Microcomputer Power, Ithaca, NY, USA) to evaluate multivariate relationships. Correlation matrices were visualized through heatmaps constructed using the Chiplot online platform (https://www.chiplot.online/ accessed on 24 August 2025), with color gradients representing correlation strength and asterisks indicating statistical significance.

## Figures and Tables

**Figure 1 plants-14-03194-f001:**
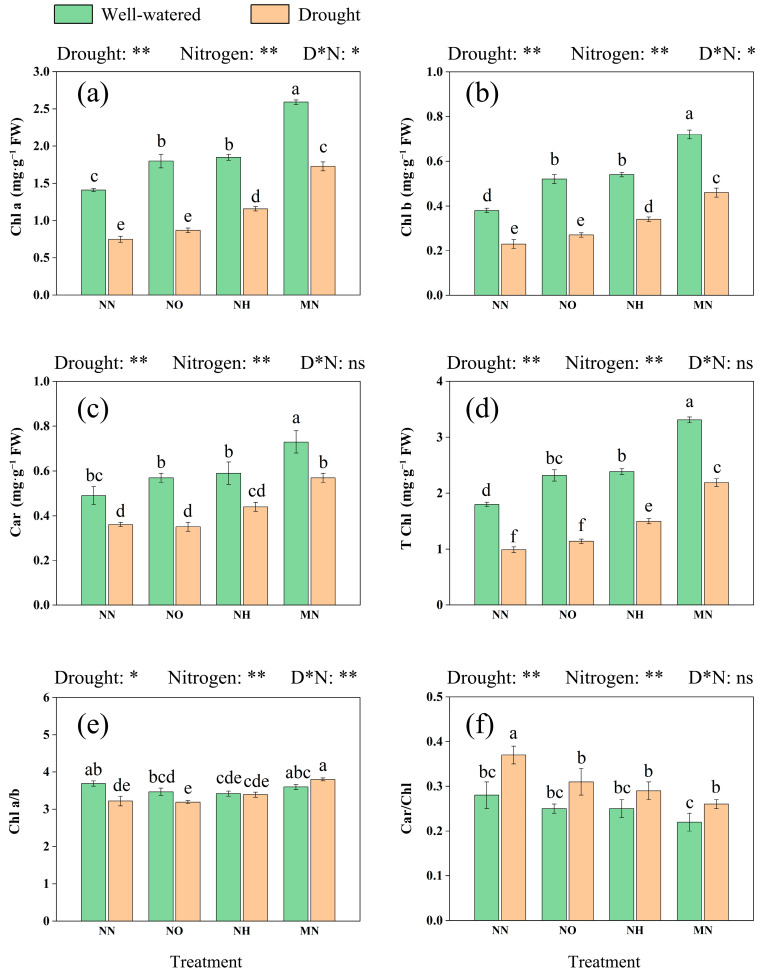
Drought and nitrogen effects on photosynthetic pigment in leaves of *M. yunnanensis* seedlings. (**a**) Chl a: chlorophyll a; (**b**) Chl b: chlorophyll b; (**c**) Car: carotenoids; (**d**) T Chl: total chlorophyll; (**e**) Chl a/b: The ratio of chlorophyll a to b; (**f**) Car/Chl: The ratio of chlorophyll to carotenoid. Note: D* N: Drought* Nitrogen. *: *p* < 0.05; **: *p* < 0.01; ns: *p* > 0.05. Different lowercase letters indicate significance at *p* < 0.05 level.

**Figure 2 plants-14-03194-f002:**
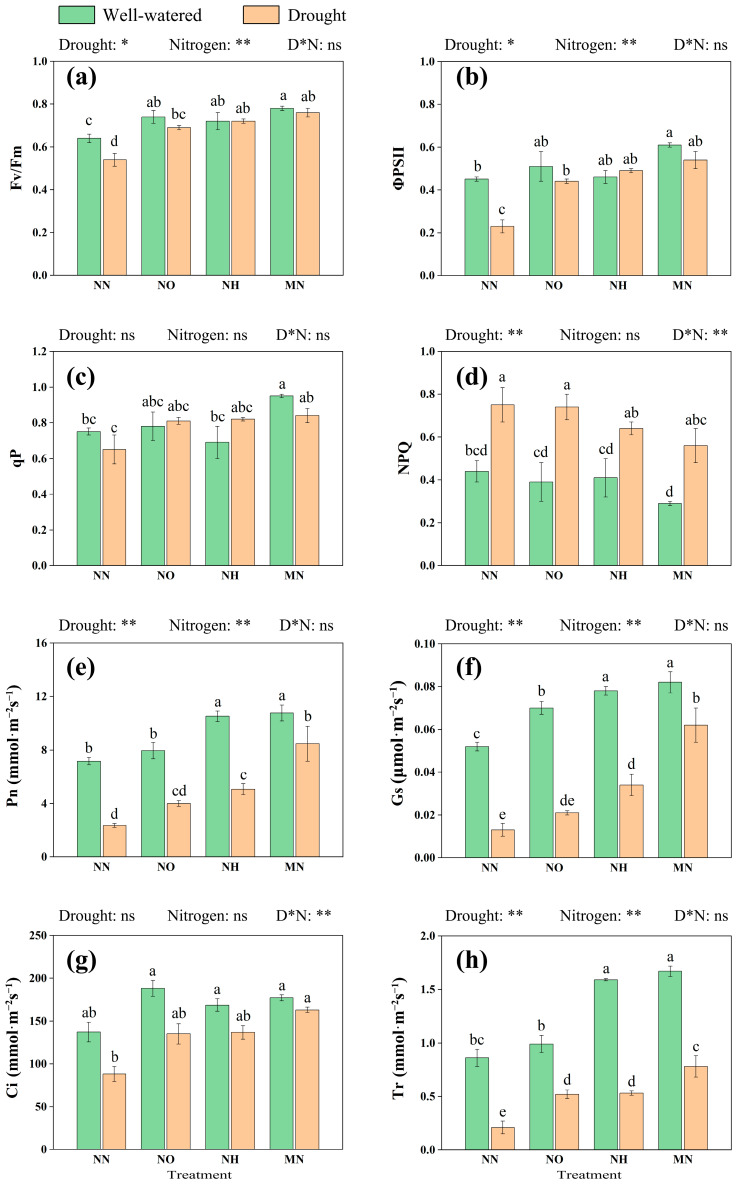
Drought and nitrogen effects on chlorophyll fluorescence and photosynthetic parameters in *M. yunnanensis* seedlings. (**a**) Fv/Fm: maximum quantum efficiency of photosystem II; (**b**) ΦPSII: photosystem II; (**c**) qP: photochemical quenching; (**d**) NPQ: non-photochemical quenching; (**e**) Pn: net photosynthetic rate; (**f**) Gs: stomatal conductance; (**g**) Ci: intercellular CO_2_ concentration; (**h**) Tr: transpiration rate. Note: D* N: Drought* Nitrogen. *: *p* < 0.05; **: *p* < 0.01; ns: *p* > 0.05. Different lowercase letters indicate significance at *p* < 0.05 level.

**Figure 3 plants-14-03194-f003:**
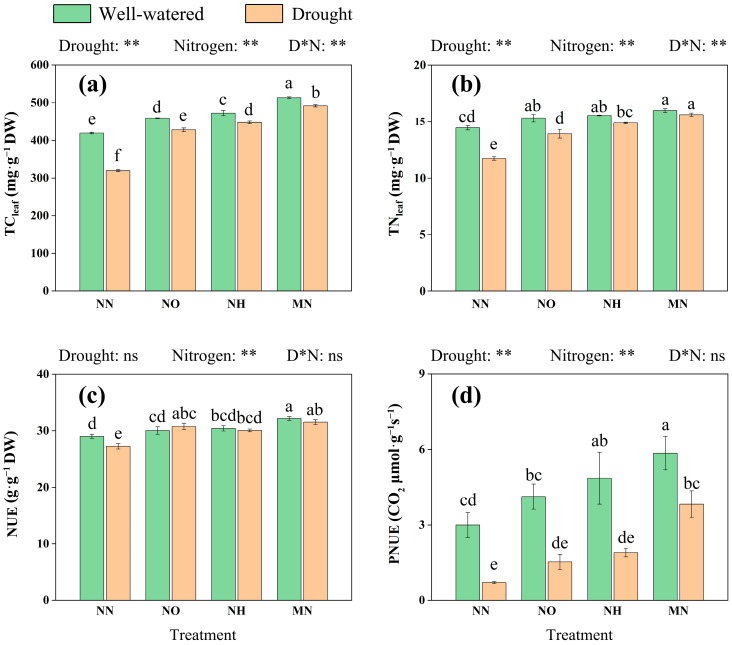
Drought and nitrogen effects on leaf nitrogen use efficiency in *M. yunnanensis* seedlings. (**a**) TC_leaf_: total carbon in leaf; (**b**) TN_leaf_: total nitrogen in leaf; (**c**) NUE: nitrogen use efficiency; (**d**) PNUE: photosynthetic nitrogen use efficiency. Note: D* N: Drought* Nitrogen. **: *p* < 0.01; ns: *p* > 0.05. Different lowercase letters indicate significance at *p* < 0.05 level.

**Figure 4 plants-14-03194-f004:**
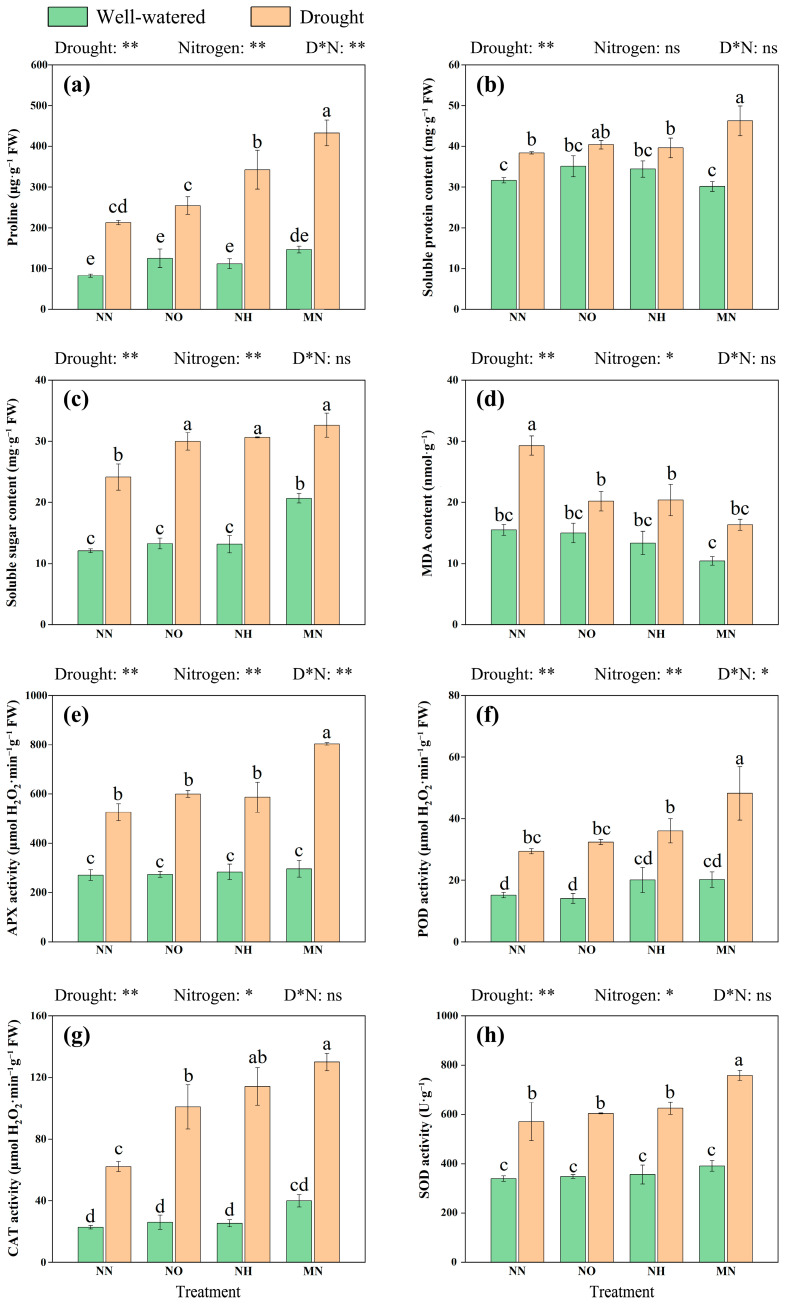
Drought and nitrogen effects on osmolytes and antioxidant capacity in *M. yunnanensis* seedlings. (**a**) Proline; (**b**) Soluble protein content; (**c**) Soluble sugar content; (**d**) MDA content: Malondialdehyde content; (**e**) APX: ascorbate peroxidase; (**f**) POD: peroxidase; (**g**) CAT: catalase; (**h**) SOD: superoxide dismutase. Note: D* N: Drought* Nitrogen. *: *p* < 0.05; **: *p* < 0.01; ns: *p* > 0.05. Different lowercase letters indicate significance at *p* < 0.05 level.

**Figure 5 plants-14-03194-f005:**
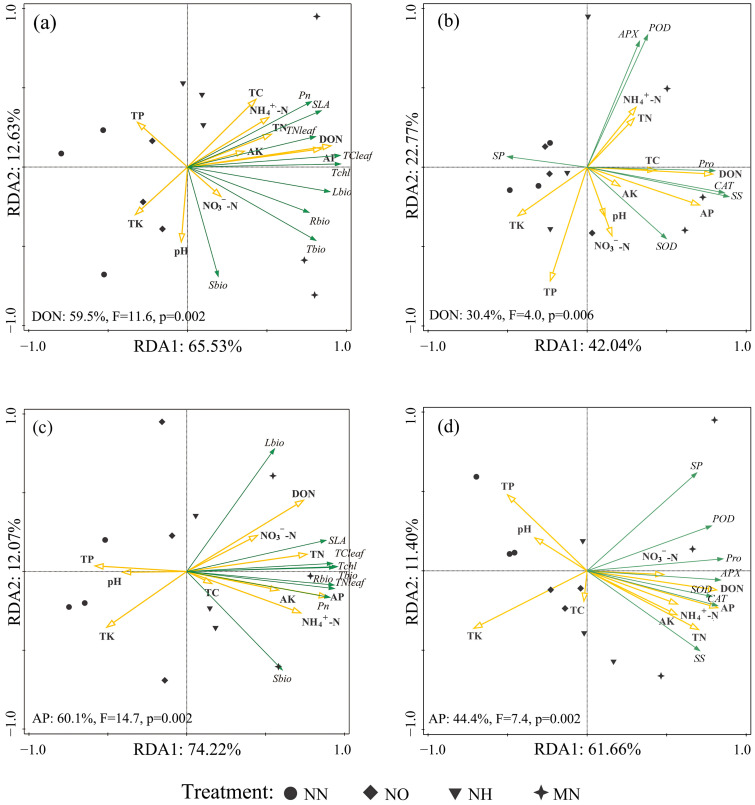
Relationships between soil chemistry and *M. yunnanensis* seedling growth–drought resistance. Note: (**a**) RDA of the effects of soil chemical properties on the growth and photosynthesis of *M. yunnanensis* seedlings under well-watered conditions; (**b**) RDA of the effects of soil chemical properties on the indexes of stress resistance of *M. yunnanensis* seedlings under well-watered conditions; (**c**) RDA of the effects of soil chemical properties on the growth and photosynthesis of *M. yunnanensis* seedlings under drought stress; (**d**) RDA of the effects of soil chemical properties on the indexes of stress resistance of *M. yunnanensis* seedlings under drought stress.

**Figure 6 plants-14-03194-f006:**
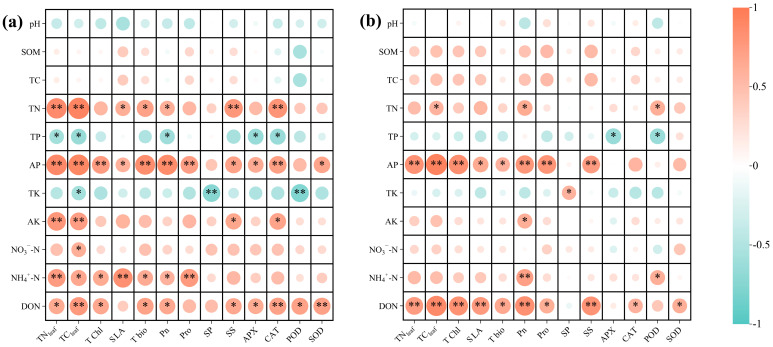
Correlation heatmap between physiological characteristics of *M. yunnanensis* seedlings and rhizosphere chemical properties. Note: (**a**): correlation between physiological characteristics and rhizosphere chemical properties under well-watered conditions; (**b**): correlation between physiological characteristics and rhizosphere chemical properties under drought stress; *: *p* < 0.05; **: *p* < 0.01.

**Table 1 plants-14-03194-t001:** Effects of drought and nitrogen on physicochemical properties of rhizosphere soil of *M. yunnanensis* seedlings.

Feild Capacity	Nitrogen From	pH	TC (g/kg DW)	SOM (g/kg DW)	TN (g/kg DW)	TP (g/kg DW)	AP (mg/kg DW)	TK (g/kg DW)
W	NN	5.67 ± 0.07bc	150.23 ± 1.45a	258.99 ± 2.50a	5.89 ± 0.02d	1.61 ± 0.02ab	130.16 ± 0.47b	13.74 ± 0.33a
W	NO	5.97 ± 0.02a	149.89 ± 1.68a	258.42 ± 2.90a	6.23 ± 0.04cd	1.55 ± 0.03b	138.55 ± 0.13a	13.91 ± 0.56a
W	NH	5.28 ± 0.01d	153.42 ± 1.88a	264.50 ± 3.24a	6.66 ± 0.31ab	1.58 ± 0.03ab	140.59 ± 3.21a	13.16 ± 0.39a
W	MN	5.75 ± 0.01b	153.65 ± 1.49a	264.88 ± 2.58a	6.39 ± 0.04bc	1.56 ± 0.04ab	145.37 ± 0.23a	13.30 ± 0.06a
D	NN	5.53 ± 0.08c	138.68 ± 0.61b	239.10 ± 1.06b	5.46 ± 0.03e	1.66 ± 0.01a	91.75 ± 0.35d	14.21 ± 0.27a
D	NO	5.78 ± 0.11b	141.29 ± 2.75b	243.58 ± 4.74b	6.80 ± 0.04a	1.57 ± 0.01ab	109.88 ± 1.70c	13.77 ± 0.08a
D	NH	4.99 ± 0.01e	140.59 ± 2.66b	242.38 ± 4.59b	6.66 ± 0.03ab	1.59 ± 0.06ab	111.39 ± 3.71c	13.69 ± 0.08a
D	MN	5.33 ± 0.05d	139.92 ± 5.78b	241.23 ± 9.96b	6.73 ± 0.02ab	1.56 ± 0.02ab	122.43 ± 5.45b	13.25 ± 0.59a
Sig	D	ns	**	**	ns	ns	**	ns
	N	**	ns	ns	**	ns	**	ns
	D* N	ns	ns	ns	**	ns	ns	ns
**Feild Capacity**	**Nitrogen From**	**NO_3_^−^-N** **(mg/kg DW)**	**NH_4_^+^-N** **(mg/kg DW)**	**AK** **(mg/kg DW)**	**DON** **(g/kg DW)**	**TC/TN**	**TC/TP**	**TN/TP**
W	NN	323.22 ± 2.12c	12.72 ± 0.71de	2079.00 ± 5.40c	591.56 ± 5.24c	25.50 ± 0.21a	93.16 ± 1.45abc	3.65 ± 0.03b
W	NO	420.94 ± 0.32a	13.72 ± 0.09d	2169.42 ± 31.39bc	598.12 ± 9.30c	24.06 ± 0.20ab	96.31 ± 2.62ab	4.00 ± 0.09ab
W	NH	333.29 ± 32.73c	25.71 ± 1.39a	2284.75 ± 43.05a	640.35 ± 8.23b	23.16 ± 1.42b	97.13 ± 2.32ab	4.22 ± 0.23a
W	MN	373.35 ± 3.34b	19.13 ± 0.46b	2190.42 ± 45.28ab	667.62 ± 6.96a	24.03 ± 0.28ab	98.28 ± 1.86a	4.09 ± 0.12a
D	NN	397.69 ± 0.53ab	10.31 ± 0.46e	1786.42 ± 1.48d	330.72 ± 8.93e	25.39 ± 0.12a	83.55 ± 0.31d	3.29 ± 0.00c
D	NO	423.23 ± 0.14a	10.95 ± 0.47e	1847.50 ± 23.29d	351.76 ± 7.78cd	20.79 ± 0.50c	90.09 ± 1.39bcd	4.33 ± 0.06a
D	NH	402.90 ± 0.96ab	16.82 ± 1.28bc	1860.33 ± 2.30d	351.18 ± 5.24cd	21.12 ± 0.50c	88.42 ± 2.96cd	4.19 ± 0.16a
D	MN	415.81 ± 2.05a	15.20 ± 0.80cd	1841.67 ± 10.18d	367.07 ± 5.55d	20.79 ± 0.87c	89.62 ± 3.65bcd	4.31 ± 0.06a
Sig	D	ns	*	**	**	**	**	ns
	N	ns	ns	ns	ns	**	ns	**
	D* N	ns	ns	ns	ns	ns	ns	*

Note: Sig: Significance; D: Drought; N: Nitrogen; D* N: Drought* Nitrogen. *: *p* < 0.05; **: *p* < 0.01; ns: *p* > 0.05. Different lowercase letters indicate significance at *p* < 0.05 level. TC: total carbon; TN: total nitrogen; TP: total phosphorus; DON: dissolved organic nitrogen; NO_3_^−^-N: nitrate nitrogen; NH_4_^+^-N: ammonium nitrogen; AP: available phosphorus; TK: total potassium; AK: available potassium; SOM: soil organic matter.

**Table 2 plants-14-03194-t002:** Drought and nitrogen effects on growth morphology of *M. yunnanensis* seedlings.

Field Capacity	Nitrogen From	Leaf Length-Width Ratio	Single Leaf Area (cm^2^)	Total Leaf Area (cm^2^)	Specific Leaf Area (cm^2^)	Plants Height (cm)	Root Length (cm)
W	NN	1.65 ± 0.08bc	4.76 ± 0.14bc	33.45 ± 3.64bc	89.73 ± 5.08c	10.37 ± 0.38de	10.90 ± 1.02b
W	NO	1.31 ± 0.01d	5.05 ± 0.15abc	41.98 ± 3.03ab	89.44 ± 2.78c	12.87 ± 0.82ab	11.87 ± 0.98ab
W	NH	1.38 ± 0.02d	5.16 ± 0.13ab	38.14 ± 7.94abc	103.13 ± 2.67ab	13.33 ± 0.33ab	13.50 ± 0.87a
W	MN	1.36 ± 0.02d	5.63 ± 0.14a	48.61 ± 2.77a	112.53 ± 2.78a	14.27 ± 0.24a	14.37 ± 0.30a
D	NN	1.81 ± 0.06a	3.55 ± 0.19d	18.83 ± 0.67d	63.06 ± 4.67d	9.27 ± 0.18e	12.10 ± 0.59ab
D	NO	1.69 ± 0.05abc	4.32 ± 0.21c	27.43 ± 2.30cd	72.06 ± 3.57d	12.00 ± 0.29bc	12.17 ± 0.85ab
D	NH	1.77 ± 0.05ab	5.33 ± 0.47ab	32.49 ± 5.92bc	89.25 ± 4.00c	10.90 ± 0.86cd	13.90 ± 0.49a
D	MN	1.60 ± 0.02c	4.81 ± 0.33bc	39.84 ± 1.27a	96.13 ± 6.55bc	12.97 ± 0.52ab	12.97 ± 0.52ab
Sig	D	**	**	**	**	**	ns
	N	**	**	**	**	**	**
	D* N	*	ns	ns	ns	ns	ns
**Field Capacity**	**Nitrogen From**	**Leaf biomass** **(g)**	**Stem biomass** **(g)**	**Root biomass** **(g)**	**Total biomass** **(g)**	**Root/Shoot** **ratio**	**LRWC** **(%)**
W	NN	0.29 ± 0.01b	0.27 ± 0.01ab	0.25 ± 0.01c	0.81 ± 0.01bc	0.45 ± 0.02b	0.89 ± 0.02a
W	NO	0.30 ± 0.01b	0.27 ± 0.01ab	0.27 ± 0.01bc	0.85 ± 0.01b	0.47 ± 0.02b	0.91 ± 0.01a
W	NH	0.30 ± 0.01b	0.27 ± 0.01ab	0.27 ± 0.01bc	0.84 ± 0.02b	0.47 ± 0.02b	0.92 ± 0.01a
W	MN	0.36 ± 0.01a	0.28 ± 0.03a	0.32 ± 0.02a	0.96 ± 0.04a	0.49 ± 0.02b	0.93 ± 0.01a
D	NN	0.21 ± 0.01c	0.23 ± 0.01b	0.27 ± 0.00bc	0.71 ± 0.01d	0.60 ± 0.02a	0.64 ± 0.06c
D	NO	0.23 ± 0.01c	0.24 ± 0.02ab	0.28 ± 0.00b	0.76 ± 0.01cd	0.60 ± 0.01a	0.69 ± 0.05bc
D	NH	0.23 ± 0.01c	0.25 ± 0.01ab	0.29 ± 0.01ab	0.77 ± 0.00cd	0.61 ± 0.02a	0.72 ± 0.03bc
D	MN	0.24 ± 0.01c	0.26 ± 0.01ab	0.31 ± 0.00a	0.81 ± 0.01bc	0.63 ± 0.02a	0.77 ± 0.06b
Sig	D	**	*	ns	**	**	**
	N	**	ns	**	**	ns	ns
	D* N	ns	ns	ns	ns	ns	ns

Note: Sig: Significance; D: Drought; N: Nitrogen; D* N: Drought* Nitrogen. *: *p* < 0.05; **: *p* < 0.01; ns: *p* > 0.05. Different lowercase letters indicate significance at *p* < 0.05 level. LRWC: Relative water in leaf.

## Data Availability

The original contributions presented in this study are included in the article. Further inquiries can be directed to the corresponding author.

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
