# Peer review of "Synergistic Effects of Nitrogen Application Enhance Drought Resistance in Machilus yunnanensis Seedlings"

_plants, 2025, doi:10.3390/plants14203194_

Round 1

Reviewer 1 Report

Comments and Suggestions for Authors

The manuscript entitled
Synergistic Effects of Nitrogen Applying Enhance Drought Resistance in Machilus yunnanensis Seedlings, focused on resistance of
ornamental seedlings (Machilus yunnanensis Seedlings ) to drought stress after nitrogen application. The research included 2 types of nitrogen (NO: nitrate nitrogen; NH: ammonium nitrogen) and mixed types (MN: mixed nitrate–ammonium nitrogen) to
Machilus yunnanensis plants growing under drought stress conditions.

The results indicated that nitrogen supplementation, particularly mixed nitrogen application, enhances drought resistance of M. yunnanensis. It is important due to climate change and growing water scarcity problems.

Introduction: Relates well to the research topic, including actual literature.

Methodology includes a huge amount of analysis performed. 

Abstract: somewhat convoluted could have been improved.

Minor comments for autors:

Line 7: double term ‘Correspondence’

Line 17: delete double zero after dot

Line 27:  ‘and photosynthetic nitrogen use efficiency by 439.44%,’ Does the amount 439.44%, is proper? Please check it.

Line 523: remove the dot after bracket

Line 577: Start sentence with capital letter

Line 173-174 – Please insert the small letter (a-f) in the frames of figure 3

Please explain for the below question:

What can be the novelty of this research compared to other works ?

Author Response

Dear Professor: 

We would like to express our sincere gratitude for taking the time to review this manuscript and provide such in-depth and insightful comments. The following modifications have been made to the manuscript "Synergistic Effects of Nitrogen Applying Enhance Drought Resistance in Machilus yunnanensis Seedlings":

  1. Comments1: The double term of “correspondence”

Response 1: We have confirmed the removal of redundant words from our manuscript, which may have been caused by a file display error.

  1. Comments 2:About “the44% increased in MN treatment, PNUE” 

Response 2: We confirmed that value has been accurately calculated and it is derived from multiple indicators, the values for the NN treatment and MN treatment were 0.71 and 3.83 respectively (drought conditions), representing a confirmed increase of 439.44%. We adjusted the words order to make the description clearer and avoid misunderstanding of the description.

  1. Comments 3:Line 523, remove the dot after bracket “ammonium nitrogen (NH), and mixed”; Comments 4: Deleted double zero after dot in line 17; Comments 5: Line 577: Start sentence with capital letter; Comments 6: Figure 3 is missing lowercase letters.

Response 3-6: The corrections have been completed.

  1. Comments 7:About the question: “What can be the novelty of this research compared to other works ?”

Response 7: This manuscript through redundancy analysis and correlation analysis links the changes in rhizosphere soil chemical properties with the physiological and biochemical processes inside the plant. This further corroborates previous researchers' findings that nitrogen application enhances plant drought tolerance. It explains why the mixed nitrogen application method better than other forms and, accordingly, offers a cost-effective solution for drought management in Machilus yunnanensis seedlings in Yunnan Province.

Once again, we express our sincere gratitude for your careful review and insightful comments. We believe that these revisions have greatly improved the quality of this article, and we hope that our manuscript is now suitable for publication. Thank you for considering our work.

Sincerely,

Jiawei Zhou

&Xiaoxia Huang

Email: huangxx@swfu.edu.cn

Oct. 5, 2025

Reviewer 2 Report

Comments and Suggestions for Authors

I suggest, if you would like, create independent table for the effect of N, drought and their interaction. This table will help the readers see the effect of different variables altogether. 

Comments on the Quality of English Language

English is fine, but presentation could be better.

Author Response

Dear Professor:

We would like to express our sincere gratitude for taking the time to review this manuscript and provide such in-depth and insightful comments. The following modifications have been made to the manuscript "Synergistic Effects of Nitrogen Applying Enhance Drought Resistance in Machilus yunnanensis Seedlings":

First, we corrected all your annotation in this manuscript.

  1. Comments1: Title “Nitrogen Applying” change into “Nitrogen Application”

Response 1: Title has been corrected.

  1. Comments2: Regarding the raised water acidity issue

Response 2: We have added a detailed description in the experimental design section; Source of irrigation water is EU-T High-Purity water system from OuKai Environmental Technology Co., Ltd. Nanjing. Measured pH value is 6.83-7.29. The pH was measured using a Starter 2100 pH tester from Ohaus Co., USA. We have added a discussion on the impact of irrigation water on pH value. The professional insights you provided have enhanced this section. Thank you.

  1. Comments3: Regarding inaccurate and ambiguous descriptions in line 376, line 382.

Response 3: We have corrected the inaccurate description.

  1. Comments4: Line 402 “figure 5 is not enough to explanation” change to Table 2 and figure5. And the sentence “The role of mixed nitrogen sources in promoting crop growth and yield increase has been confirmed across numerous studies” was revised to “This is consistent with the findings of many studies that mixed nitrogen sources can promote the crop growth and yield.”

  1. Comments5: Line 421 about the sentence “In this study, multiple factors contributed to the decline in photosynthetic capacity.” “Is not clear”.

Response 5: We have revised it to “In this study, the decline in photosynthetic capacity may be caused by many reasons.” And the potential influencing factors were discussed in subsequent sections.

  1. Comments6: Line 433-435 “talk about stomatal closure”

Response 6: We have added more discussion about the drought effect on stomatal.

  1. Comments7: About the root length issue, how to measure.

Response 7: We have added the detailed measurement method, where the length is measured as the total root length from the main root to the tip.

  1. Comments8: Missing citation in line 560.

Response 8: We have added citation in line 560.

  1. Comments9: Create an independent table for the effect of N, drought and their interaction.

Response 9: According to your suggestion, we have created an independent table for the effect of N, drought and their interaction in appendix.

Once again, we express our sincere gratitude for your careful review and insightful comments. We believe that these revisions have greatly improved the quality of this article, and we hope that our manuscript is now suitable for publication. Thank you for considering our work.

Sincerely,

Jiawei Zhou

&Xiaoxia Huang

Email: huangxx@swfu.edu.cn

Oct. 5, 2025

Reviewer 3 Report

Comments and Suggestions for Authors

The manuscript could be an intersting paper, however it is not proper for publication in the present form.

Additionally the paper has not been prepared according to the instructions for Authors.

Results:

The results should be described according to the statistical analysis. I advise to check the whole chapter carefully. Additionally the Authors should be more careful when describing the results and avoid mental shortcuts.

There are no values presented and the standard deviation should be provided as well.

It would be better to present the values of the tested features in a table to make it more clear, instead of tens of figures. Additionally the SD should be give as well.

Discussion

It should be more clearly pointed out which data concerns the presented study and which is cited – the citations are given, however, the sentences do not always make it clear.

Materials and methods

The conditions of germination  and cultivation should be provided.

The producers of all soil or substances should be provided.

10 plants per treatment is extremely few, especially that the plant seems to be easily available. It is advised to have at least 30 samples to conduct the ANOVA. Was the normality distribution done?

How many repetitions were conducted?

The methods applied should be described more precisely.

References

The references in the manuscript should be numbered not presented in the form of names.  The references are prepared in a wrong way as well.

More comments in the manuscript.

Comments on the Quality of English Language

I would advise to have it checked by a professional translator, there are many long and difficult to understand sentences, with the vocabulary which might suggest the use of AI translation tool.

Author Response

Dear Professor: 

We would like to express our sincere gratitude for taking the time to review this manuscript and provide such in-depth and insightful comments. The following modifications have been made to the manuscript "Synergistic Effects of Nitrogen Applying Enhance Drought Resistance in Machilus yunnanensis Seedlings":

  1. Comments1: Regarding the issue of the manuscript's title being overly complex and difficult to understand.

Response 1: We have referred to the feedback from all reviewers and intend to change the title to “Synergistic Effects of Nitrogen Application Enhance Drought Resistance in Machilus yunnanensis seedlings”, we hope this title can summarize the key study points of this manuscript clearly and explicitly.”

  1. Comments2: Regarding your concern that the excessive number of figures may hinder readability and comparison for readers.

Response 2: We have changed Figure1 & 2 into tables, this should facilitate reader comprehension and data comparison.

  1. Comments3: Regarding your concern about the limited sample size.

Response 3: We are sorry for making a mistake in the sample size and number in the previous manuscript.

Actually, we set more than 30 pots for every treatment in order to select the consistent growth plants. In the revised manuscript, we have modified the sentence “with 10 pots per treatment” to “with 3 replicates per treatment and 10 pots as a replicate”.

  1. Comments4: Regarding the issue you raised about the poorly phrased passages in the text that have led to misinterpretation, such as line 104, 114.

Response 4: We have made corrections and additions to address the insufficient descriptions of the different moisture levels and treatment groups. The revised version will reduce ambiguity in the description.

  1. Comments5: Regarding your concern about the lack of detailed descriptions for several experimental methods in the Materials and Methods section., such as 4.2. Experimental design, 4.3. Nitrogen application, 4.5. Growth Parameters, 4.10. Statistical analysis.

Response 5: We have provided the necessary details and hope it can be more clearly in the revised manuscript.

Comments 6: Regarding your revisions of the multiple grammatical errors in this manuscript.

Response 6: We have adopted and corrected grammatical expressions, hoping to improve reading fluency.

  1. Comments7: With regard to the improperly formatted citations in this manuscript that you pointed out.

Response 7: We have modified this section according to the rule of the Plants journal.

Once again, we express our sincere gratitude for your careful review and insightful comments. We believe that these revisions have greatly improved the quality of this article, and we hope that our manuscript is now suitable for publication. Thank you for considering our work.

Sincerely,

Jiawei Zhou

&Xiaoxia Huang

Email: huangxx@swfu.edu.cn

Oct. 5, 2025

Round 2

Reviewer 3 Report

Comments and Suggestions for Authors

Dear Authors,

Thank you very much for the answers and changes applied. The tables allow to see the correlations easier.